# New Insights into the Implication of Epigenetic Alterations in the EMT of Triple Negative Breast Cancer

**DOI:** 10.3390/cancers11040559

**Published:** 2019-04-18

**Authors:** Noura Khaled, Yannick Bidet

**Affiliations:** Laboratoire d’Oncologie Moléculaire, Centre Jean PERRIN et IMoST, UMR 1240, Inserm/Université Clermont Auvergne 58 rue Montalembert, 63000 Clermont-Ferrand, France; yannick.bidet@uca.fr

**Keywords:** triple negative breast cancer TNBC, epithelial-mesenchymal transition EMT, epigenetic modifications, long non-coding RNAs LncRNAs, miRNAs, histone modifications, DNA methylation, metastasis

## Abstract

Breast cancer is the most common cancer and leading cause of cancer death among women worldwide, encompassing a wide heterogeneity of subtypes with different clinical features. During the last two decades, the use of targeted therapies has emerged in clinical research in order to increase treatment efficiency, improve prognosis and reduce recurrence. However, the triple negative breast cancer (TNBC) subtype remains a clinical challenge, with poor prognosis since no therapeutic targets have been identified. This aggressive breast cancer entity lacks expression of oestrogen receptor (ER) and progesterone receptor (PR), and it does not overexpress human epidermal growth factor receptor 2 (HER2). The major reason for TNBC poor prognosis is early therapeutic escape from conventional treatments, leading to aggressive metastatic relapse. Metastases occur after an epithelial-mesenchymal transition EMT of epithelial cells, allowing them to break free from the primary tumour site and to colonize distant organs. Cancer-associated EMT consists not only of acquired migration and invasion ability, but involves complex and comprehensive reprogramming, including changes in metabolism, expression levels and epigenetic. Recently, many studies have considered epigenetic alterations as the primary initiator of cancer development and metastasis. This review builds a picture of the epigenetic modifications implicated in the EMT of breast cancer. It focuses on TNBC and allows comparisons with other subtypes. It emphasizes the role of the main epigenetic modifications lncRNAs, miRNAs, histone and DNA- modifications in tumour invasion and appearance of metastases. These epigenetic alterations can be considered biomarkers representing potential diagnostic and prognostic factors in order to define a global metastatic signature for TNBC.

## 1. Introduction

Although mortality associated with breast cancer has been constantly declining over the last 20 years, there were still more than 626,000 deaths worldwide in 2018 (International Agency for Research on Cancer IARC 2018 Breast Cancer Statistics). A major reason for these therapeutic failures is the wide heterogeneity of breast cancer and distant metastases. Fifteen years ago, molecular classification of these cancers paved the way for targeted therapeutic approaches taking into account the specificities of different subtypes of breast cancer [1,2].

Triple Negative Breast Cancer (TNBC) is a highly aggressive subtype, accounting for 10–20% of all diagnosed breast cancers, with no targeted therapy available due to the lack of oestrogen and progesterone receptor expression and the absence of HER2 amplification [3]. Representing a serious clinical therapeutic challenge, TNBC is characterized by poor overall survival, early relapse and most importantly, frequent distant metastasis [3]. A lethal hallmark of cancer, metastasis of TNBC is correlated with aberrant activation of epithelial-mesenchymal transition (EMT) [4,5]. EMT is a fundamental biological mechanism by which epithelial cells acquire a mesenchymal phenotype by gaining migratory and invasive properties and modifying cell adhesion molecules [4,5]. This differentiation process is important during development, but inhibiting EMT by antagonizing the EMT-related mesenchymal cell markers such as αvβ3 integrin reveals its massive involvement in TNBC aggressiveness and metastasis [6]. It allows triple negative epithelial breast cancer cells to break free from their primary tumour site and eventually to colonize distant organs developing metastasis [4,5]. During the last decade, a large amount of data and knowledge were acquired on TNBC biomarkers. Chemokine C-C motif ligand 5 (CCL5), interleukin 17B, transforming growth factor β (TGF-β), placental growth factor, CXCL10 and platelet-derived growth factor receptor β (PDGFRβ) were considered as biomarkers for TNBC aggressiveness and progression. These biomolecules were involved in the recruitment of bone marrow-derived mesenchymal stem cells (BM-MSCs), which play a crucial role in multiple EMT program mechanisms (cell migration and invasion, neo-angiogenesis, promoting extracellular matrix remodelling…), towards TNBC sites [7]. Cancer-associated EMT requires complex and comprehensive reprogramming, including changes in metabolism, expression levels and epigenetic [4,5].

From a molecular point of view, EMT is characterized by the loss of E-cadherin, claudin and cytokeratin expression, and the overexpression of N-cadherin and vimentin, which are respectively epithelial and mesenchymal cell markers [5]. Partial EMT leads to cells with a mixed phenotype, called “metastable”. Several major pathways are involved in the EMT, including TGF-β, NF-κB, WNT, NOTCH, HIF1/2 and RAS-ERK1/2. Most of these interact with transcription factors implicated in the modulation of E-cadherin expression such as Snail, Slug, ZEB1, ZEB2, E47, Twist, and KLF8 [5].

As researchers continue to illustrate the effect of epithelial-mesenchymal plasticity on tumour progression and to investigate its invasion-related molecular mechanism, a clearer perception of the general epigenetic modifications in malignant cells undergoing EMT is now appearing. Nevertheless, the current state of art of the precise role of epigenetic regulators in controlling both EMT and MET processes in tumorigenesis including TNBC are not fully understood yet. In sharp contradiction with EMT inducing factors and transcription factors, we are just now beginning to understand how those alterations induce EMT and MET or function to maintain epithelial and mesenchymal phenotypes. For this reason, the exact epigenetic regulation remains a frontier to discover.

We hypothesize that close interactions between different EMT inducing factors and TFs and epigenetic regulators exist. EMT-related factors and markers are targets of epigenetic regulations. Epigenetic modifiers must also be regulated by TFs to transcriptionally regulate epithelial and mesenchymal genes during EMT. For example, TFs may be responsible for the expression of epigenetic factors activating or repressing in return the expression of EMT-markers. The understanding of these probable functional interactions controlling the inter-conversion between epithelial and mesenchymal phenotypes may provide new opportunities and novel therapeutic approaches aiming towards the prevention of cancer invasion and metastasis, based on the development of individualized epigenetic cancer therapy.

A lot of work has been published on TNBC, and mutations and modifications of expression levels are now well described. In our review, we would discuss the current state-of-the-art and latest findings regarding the implication of epigenetic modifications in EMT. We would emphasize on the role of four major epigenetic modifications in the regulations of the EMT of TNBC responsible for distant metastases, describing the various molecular pathways implicated as well as a large number of transcription factors influenced by the deregulation of these factors. They include non-coding RNA, both long-non-coding RNAs (lncRNAs) and micro RNA (miRNAs), and modifications at the level of histones or DNA, such as acetylation or methylation.

For each class of epigenetic factors (lncRNA, miRNA, histone or DNA modification), we first present large screening studies that provide lists of candidates specifically dysregulated in TNBC. These general studies pave the way for targeting TNBC through the identification of high impact candidate genes and factors. However, from one study to another, the sets of genes often overlap poorly, and functional validation is required. Thus, in the second part of each chapter, we focus on studies describing the role of epigenetic factors involved in the EMT of TNBC. The functional evidences are not paraphrased for each factor, but they are gathered in Table 1, Table 2, Table 3, Table 4 and Table 5 and a comprehensive network assembles all interactions in Appendix A. Further genes are pending for validation in order to identify novel prognostic factors and therapeutic targets. A number of essential anti-TNBC drugs used (Table 4) and a description of primary applied clinical trials are discussed in this paper.

## 2. Long Non-Coding RNAs

Long non-coding RNAs (lncRNAs) are transcribed RNA molecules not encoding any protein, with a length over 200 nucleotides [8]. Due to the lack of experimental tools and overall knowledge, the function of lncRNAs was largely unknown until the early 1990′s when H19 and Xist were found to be involved in epigenetic modulations [9]. Since the publication of the human genome in 2001 and the finding that only 2% of the genome encodes for proteins despite the fact that >90% is actively transcribed [10], extensive attention has been given to long non-coding RNAs in order to investigate their biological and regulatory activity. While considering that dysregulation of lncRNAs is well described in many human disorders, such as cardiovascular disease [10], prior investigations have shown that dysregulation of lncRNAs plays a critical role in cancer progression, as oncogenic or tumour-suppressor genes by modulating tumour cell death, proliferation, migration, invasion and drug resistance [11]. In recent years, researchers were highly interested in the role of lncRNAs in several subtypes of breast cancer including TNBC [12,13,14,15,16]. LncRNAs have been reported as promising diagnostic biomarkers, prognostic factors and potentially efficient therapeutic targets by a set of systematic research described further down. Moreover, these studies have evaluated the role of single lncRNAs for different endpoints, such as overall survival OS, progression-free survival PFS, poor prognosis, and TNBC proliferation and metastasis.

Lately, abnormal expression of lncRNAs, related altered mechanisms and primary deregulation were the subject of pivotal studies presented later.

In a microarray analysis study, a set of 1403 unique lncRNAs, with 853 up-regulated and 550 down-regulated, was discovered to be differentially expressed between TNBC and non-tumourous tissues, suggesting a pivotal role of lncRNAs in TNBC progression [17]. In the aim of offering specific therapeutic targets and promising diagnosis biomarkers, a core group of onco-lncRNAs was identified. Another set of 1211 differentially expressed TNBC-specific lncRNAs among the wide dysregulated lncRNAs was defined by co-expression networks [18].

Furthermore, a lncRNA expression signature of four lncRNAs (*IGKV*, *RP11-434D9.1*, *BC016831*, and *LINC00052*) differentially expressed between TNBC and non-TNBC tissues was validated [19]. Recently, a re-mapping strategy generated novel data allowing the discovery of additional differentially expressed lncRNAs in TNBC. Sixty one recent lncRNAs were identified, where thirty-three lncRNAs were downregulated (Top five: *lnc-PAPLN-2:1*, *lnc-FLT3LG-1:7*, *lnc-NEK8-2:1*, *lnc-FLOT2-1:1* and *lnc-ZNF75D-2:2*) and twenty eight lncRNAs were upregulated (*Top five: lnc-DNAJC16-1:1*, *lnc-SC5DL-3:1*, *lnc-PURA-2:1*, *lnc-EIF2C2-1:1* and *lnc-ELP4-3:1*) [20].

From our intention to review the recent advances in this field evaluating the crucial role of each lncRNA in a single study separately in the outcome of the ‘invasion-metastasis cascade’ in TNBC patients, a set of EMT specific lncRNAs are presented. lncRNAs involved in EMT of TNBC could be assigned to two classes based on the level of expression and molecular regulation. First, lncRNAs *LET* [21], *Xist* [22], *lncRNA-CTD-2108O9.1* [23], and *GAS5* [24] were shown to be downregulated in TNBC cell lines. Upregulation of these lncRNAs attenuates TNBC cell proliferation and viability and enhances apoptosis. In addition, the overexpression of these lncRNAs inhibits TNBC growth, invasion and migration, and represses EMT by increasing E-cadherin expression as well as decreasing N-cadherin and Vimentin expression. The mechanisms by which this category of lncRNA acts differ from one lncRNA to another. *XIST* exerts its suppressive activity via miR-155/CDX1 [22], *GAS5* by functioning as a competing endogenous RNA (ceRNA) that antagonizes miR-196a-5p targeting thereby FOXO1/PI3K/AKT signalling [24], *lncRNA-CTD-2108O9.1* targets leukemia inhibitory factor receptor LIFR [23], while no specific molecular mechanism for *LET* is yet defined [21]. (see Appendix A and Table 1).

Twenty-three different lncRNAs investigated in separated studies were reported to be pro-tumoural agents, usually correlated with a high capacity of TNBC progression and metastasis, poor overall survival and early outcome. Knockdown of these upregulated TNBC lncRNAs leads eventually to the suppression of cell motility along with TNBC metastasis and invasion and to the inhibition of EMT molecular marker expression. The mechanisms of actions of these lncRNAs, *MALAT1* [25,26,27], *HOTAIR* [28], *SNHG12* [29], *LINC01638* [30], *H19* [31], *Linc-ZNF469-3* [32], *LINC00673* [33], *lnc-ATB* [34], *HOXA11-AS* [35], *OR3A4* [36], *MIAT* [37], *GHET1* [38], *AC026904.1/UCA* [39], *NNT-AS1* [40], *PVT1* [41], *ARNILA* [42], *lncRNA-RoR* [43], *LINC00518* [44], *ADAM/lnc015192* [45], *LOC554202* [46], and *snaR* [47] are summarized in Table 1 (see Appendix A).

Among these two profiles, some lncRNAs, GAS5 [24], MIAT [37], ARNILA [42], NNT-AS1 [40], lnc015192 [45], and UCA [39] are able to directly regulate EMT pathways and TNBC invasion by functioning as ceRNAs for EMT-regulating-RNA via sharing common sequences with them (see Appendix A and Table 1).

## 3. MiRNAs

MicroRNAs (miRNAs) are small noncoding RNAs approximately 22 nucleotides long, involved in the regulation of posttranscriptional gene expression [48]. Aberrant miRNA expression has been associated with cancer cell progression, invasion and metastasis, acting both as oncogenes and tumour suppressors [49,50,51,52,53,54,55]. miRNAs are also powerful drivers of metastasis in many malignancies by regulating non-EMT-related and/or EMT–related mechanisms [56].

As miRNAs are smaller and more stable than mRNAs in formalin-fixed paraffin-embedded (FFPE) tissues and accessible body fluids, their expression was the subject of multiple investigations mentioned later, as potential TNBC biomarkers in order to give insight into their role in TNBC progression and invasion along with the appearance of distant metastasis. A number of general studies evaluating a panel of miRNAs by different technics are described lately.

Recently, using the robust rank aggregation (RRA) method, metasignatures of six significantly dysregulated miRNAs: two downregulated: *hsa-miR-449a* and *hsa-miR-190b* and four upregulated: *hsa-miR-9-5p*, *hsa-miR-18a-5p*, *hsa-miR-135b-5p* and *hsa-miR-522-3p*, were identified from different studies, with high prediction accuracy. The gene ontology analysis revealed the top 10 preferred target genes and 10 pathways of metasignature miRNAs in TNBC suggesting promising candidate for diagnostic biomarkers in TNBC [57].

Four miRNA signatures in TNBC given by *miR-155*, *miR-493*, *miR-30e* and *miR-27a* expression levels were identified allowing a new stratification of TNBC based on the level of patient outcome. Defined as “risk”-associated, down-regulation of *miR-27a* and *miR-30e* is correlated with worse patient outcome, while upregulation of *miR-493* and *miR-155* associated with better outcome is described as “protective” [58].

Furthermore, miRNAs expression profile generated by *limma* algorithm identified 67 significantly differentially expressed miRNAs (DEmiRNAs) between TNBC and normal tissues. Among them 21 were down-regulated (Top five: *hsa-mir-486-1*, *hsa-mir-486-2*, *hsa-mir-4732*, *hsa-mir-139* and *hsa-mir-451a*) and 46 up-regulated (Top five: *hsa-mir-105-1*, *hsa-mir-519a-1*, *hsa-mir-105-2*, *hsa-mir-767* and *hsa-mir-516a-1*) [59].

Further profiling of miRNAs expression in TNBC cells identified four candidate miRNAs (miR-26a, miR-153, miR-10b and miR-146a) as potential TNBC biomarkers. Exogenous expression assays reported that miR-26a and miR-10b suppress *BRCA1* expression in MDA-MB-231 and MCF-7 cells, while BRCA1 expression was downregulated by miR-153 only in MCF7 cells. High expression of miR-146a in TNBC cells compared to non-TNBC was validated by in silico analysis of The Cancer Genome Atlas (TCGA) data [60].

Seven other miRNAs (*miR-300*, *miR-382*, *miR-494*, *miR-495*, *miR-539*, *miR-543*, and *miR-544*) in the imprinted DLK1-DIO3 region are known to cooperatively repress EMT. DLK1-DIO3 miRNAs target EMT-inducing molecules repressing a signaling network comprising ZEB1/2, TWIST1, BMI1, and miR-200 family miRNAs. Thus, silencing the cluster by hypermethylation of upstream CpG islands in MDA-MB-231 cells represents a key early step that promotes loss of CDH1 expression, initiates EMT, and activates tumour cell invasion and metastasis [61]. Three main members (*miR-200 a*, *b* and *c*) of the miR200 family are considered as tumour suppressor miRNAs restraining EMT. These miRNAs targeting E-cadherin suppressors ZEB1, ZEB2 and SUZ12 are downregulated in TNBC cells, representing a promising therapeutic target [62,63,64] (see Appendix A and Table 2).

Given that many lncRNAs act as competing endogenous lncRNA (ceRNA) to sequester miRNAs, and since miRNAs rank among the main drivers of EMT in many malignancies, this report aims to generate a general summary of all miRNAs differentially expressed in TNBC, as promising new therapeutic targets and novel molecular biomarkers for TNBC. Several data suggest that chromatin modifications are correlated with differential expression of lncRNAs and miRNAs in TNBC. Promoter hypermethylation was shown to be one of the major mechanisms to silence miR-200c/miR-141 and LOC554202/miR-31 locus [46,64]. It is very likely that the same regulation occurs with other lncRNAs and miRNAs involved in the EMT of TNBC.

Among the numerous miRNAs identified by these large screening assays, single studies aiming to investigate the precise performance of some specific miRNAs in the EMT of TNBC were proceeded. The goal was to determine the level of miRNAs expression in TNBC highly metastatic cells in vitro and to alter this expression. The analysis of epithelial and mesenchymal markers was used to determine the anti- or pro-metastatic role of the candidate miRNAs. For a number of these miRNAs, target genes were predicted and tested in order to identify the specific mechanism of action of the miRNAs.

A set of twenty two miRNAs correlated with EMT and TNBC cell migration and invasion suggesting a leading role in influencing the expression levels of their target genes and pathways can be established: *miR-30a* [65], *miR-146a* [66], *miR-143-3p* [67], *miR-200b-3p/miR-429-5p* [68], *miR-139-5p* [69], *miR-212-5p* [70], *miR-199a-5p* [71], *miR-155* [72], *miR-34c-3p* [73], *miR-3178* [74], *miRNA-200b-3p* [75], *miR-125b* [76], and *miR-655* [77], were outlined as down regulated while *miR27-a* [78], *miR-182* [79,80], *miR-454* [81], *miR-373* [82], *miR-221/222* [83], and *miR-10b* [84] were reported as up regulated (see Appendix A and Table 2).

## 4. Histone Modifications

N-terminal modifications of histones, including acetylation and methylation, as well as the reverse of these processes frequently observed in cancer, are responsible for modifying the secondary DNA structure and the accessibility of transcription factors to gene promoter regions regulating gene transcription [85,86,87]. In breast cancer, histones modifications play a major role in cell proliferation, induced apoptosis and metastatic ability [88].

Discussing histone modifications mechanisms in TNBC especially, surveys proposed an interesting performance for histones modifications in the appearance and progression of TNBC [89,90,91]. Despite the not fully understood effect of histones alterations, histone modification inhibitors, such as histone deacetylase inhibitors (HDACis) have emerged as a promising new class of multifunctional anticancer agents preventing TNBC growth and metastasis [92]. These inhibitors could also help to better understand the effect of histone modifications on the outcome and the metastasis of TNBC through gene expression regulation in order to define new biomarkers and targeted therapies [92,93]. Nonetheless, two HDACis used together with ionizing radiation, autophagy and kinase inhibitors were the subject of multiple investigations [94,95,96].

Identifying major histone alteration’s delineation in TNBC as well as principle genes influenced by these modifications is the subject of upcoming surveys. A specific histone modification profile identified eight key histone modifications (*H3K9me3*, *H3K9ac*, *H3K27ac*, *H3K4me3*, *H3K27me3*, *H3K4me1*, *H3K36me3* and *H3K79me2*) in four different TNBC cell *lines* (*MDA-MB-231*, *MDA-MB-468*, *MDA-MB-436*, and *HCC1937*) among thirteen cell lines [97]. Intriguingly, a TNBC specific gene was also identified: Actin Filament Associated Protein Antisense RNA 1 (AFAP1-AS1), an anti-sense lncRNA marked by numerous active histone modifications, such as H3K79me2 and H3K4me3, suggesting a role in tumour EMT and progression. The same study showed that a new histone acetyltransferase NAA60, which mediates acetylation events in H4 such as H4K20ac, H4K79ac and H4K91ac, was silenced in TNBC [97]. Furthermore, a global map of histone H3 methylation (H3K4me3) and acetylation (H3K4ac) profiles in MDA-MB-231 cells showed an association of H3K4me3 with late-stage cancer cells, while gain of H3K4ac was correlated with cancer-related phenotypic traits, especially EMT pathways [98].

Histone methyltransferases (HMTs) catalyse the methylation of lysine and arginine residues. Genetic modifications of HMTs affect tumour progression and metastasis. Four HMTs (*ASH1L*, *SETDB1*, and *SMYD3*) presented higher amplification frequencies in basal-like breast cancer (a subtype of TNBC) while eight HMT expression levels (*EZH1*, *SMYD3*, *EHMT1*, *SETD7*, *PRDM4*, *SETD3*, *SETD1B*, and *PRDM6*) were significantly downregulated and twelve HMT expression levels (*EZH2*, *PRDM15*, *PRDM13*, *SMYD2*, *SMYD5*, *SUV39H1*, *SUV39H2*, *EHMT2*, *WHSC1*, *SETD8*, *SETDB* and *SETD6*) were significantly up-regulated comparing basal-like to other breast cancer subtypes [99].

A multi-parametric RNAi screening approach identified 70 gene candidates involved in EMT. Among them, three histone alteration modulators, histone methyltransferase G9a, H3K79 methylator *DOT1L* and a new histone acetyltransferase *KAT5*, were shown to be up-regulated, repressing CDH1 and enhancing EMT [100].

DNMTi SGI and HDACi MS275 have been proposed as combined treatment for TNBC able to epigenetically reverse EMT and inhibit cell motility, proliferation and colony formation. This anti-EMT/antitumour activity is exerted by impeding EpCAM cleavage and the WNT pathway, repressing abnormal p53, EZH2, and ZEB1, and by promoting E-cadherin and apoptosis, together with histone H3 tri-methylation [101].

DNA damage-specific histone chaperone aprataxin PNK-like factor (APLF), significantly enhanced in TNBC, was reported as an essential TNBC biomarker and a treatment target. Downregulation of APLF upregulates E-cadherin (CDH1) expression and suppresses EMT related genes inhibiting MDA-MB-231 cell invasion and metastasis [102].

Based on histone modifications profiling, several studies aimed to reconstitute the normal expression level of epigenetically deregulated genes. For this reason, metastatic TNBC cells have been treated with specific histone modifiers inhibitors to investigate their ability to induce or reduce TNBC metastasis and to evaluate their global effect on the TNBC EMT. Furthermore, the effect of depletion of upregulated histones modifiers on migration and invasive characteristics has been investigated in TNBC metastatic cells. Variations of epithelial and cell adhesion markers were also analysed in order to understand the roles of histone modifications on EMT and, eventually, identify potential diagnostic and prognostic biomarkers.

The role of multiple histone alteration effectors in inducing or reversing EMT in TNBC are listed in Table 3 (see Appendix A). EMT-repressive factors *TIP 60* [103] and *KDM6A* [104] and EMT-inductive factors *EZH2* [105], *LSD1* [106], *hSETD1A* [107], *JMJD5* [108], *KDM5B* [109], *KDM3A* [110] and *HDAC8* [111] have been investigated in multiple studies (Table 3 and see Appendix A). Histone modification inhibitors used in studies evaluating the role of specific histone modification treatments in promoting or inhibiting the EMT of TNBC are listed in a supplementary table (Table 4 and see Appendix A). The mechanisms of treatment of the histone deacetylase inhibitors vorinostat (*SAHA*) [94,112], panobinostat (*LBH589*) [93], entinostat (*ENT*) [113,114], trichostatin A (*TSA*) [115] and *HTPB* [116], as well as the lysine demethylase inhibitor parnate (*PCPA*) [117], and the histone methyltransferase inhibitor *UNC0638* [118], along with their main role in reversing EMT are listed in Table 4 (see Appendix A).

## 5. DNA Methylation

DNA methylation is the first and the best-described epigenetic modification in tumours characterized by abnormal activity of DNA methyltransferase enzymes (DNMTs), principally DNMT1 [118]. Two main characteristics delineate the cancer epigenome; gene-specific hyper-methylation and global DNA hypo-methylation [119]. Gene-specific hyper methylation of caretaker and tumour suppressor genes is caused by cytosine methylation in CpG islands, which is a recognized epigenetic silencing marker and a key factor in breast carcinogenesis [119,120,121]. The identification of differentially methylated regions (DMRs) enriched with promoters associated with DNA hypersensitive sites and transcription factor binding sites can be used to sort TNBCs into three different methylation clusters with different prognosis. Three DMRs were associated with good survival, while fourteen DMRs were associated with poor survival [122].

A diverse set of DNA methylation signatures predicting TNBC treatment response, identifying different breast cancer subtypes with significant diagnostic and prognostic potential have been presented by many studies.

First, an association is strongly suggested between the loss of expression of epithelial biomarkers and gene promoter hypermethylation events in human TNBC. Comparison of the hypermethylation of 110 CpGI in 69 cancer-related genes between TNBC and non-TNBC revealed a similar number of methylated CpGI but a different list of methylated genes [123].

A specific methylation profile of sixteen cancer-related genes specific to TNBC was defined by the non-methylation of eleven genes (*MSH2*, *MLH1*, *MSH3*, *MSH6*, *CACNA1A*, *CACNA1G*, *GSTP1*, *ID4*, *PMS2*, *DLC1*, and *TWIST1*) and the methylation of five genes (*RB*, *CD44*, *CDKN2B*, *MGMT*, and *p73*) [123].

Among 38 genes identified as deregulated in TNBC, whole genome DNA methylation analysis revealed an essential role of DNA methylation in modifying the expression of 16 TNBC-specific genes expression. Only *IL6ST* showed significantly altered methylation in three probes allowing it to be classed as a DMR [124].

The analysis of EMT-related marker expression in TNBC described a new signature for TNBC subclasses, based on the association of CD146 or MCAM expression (an EMT inducer) with the expression of EMT and breast cancer stem cell (BCSC) (CD44+/CD24−) markers, and suggested that ZEB1 expression as a promising biomarker for poor clinical outcome [125].

Aside from DNA methylation inhibition of anti-tumour genes, hypermethylation or hypomethylation of miRNA were responsible for main miRNAs deregulation mentioned above. Addition to the pro-tumour miRNAs up-regulated in TNBC by the absence of methylation, a set of miRNAs displaying anti-tumour characteristics were epigenetically inactivated by DNA methylation (*DLK1-DIO3 cluster*, *miR126*, *miR195*, *miR497*, *miR148a*, *miR152* and *miR-200 family*) [126].

A TNBC gene expression signature has been linked to aberrant promoter CpG hypermethylation of nine epigenetic biomarker genes (*CDH1*, *CEACAM6*, *CST6*, *ESR1*, *GNA11*, *MUC1*, *MYB*, *SCNN1A*, and *TFF3*). The loss or reduced level of methylation-sensitive gene products was defined as a fundamental characteristic of TNBC, and associated with poor long-term survival [127].

Several genes chosen from the DNA-methylation-based profiles were evaluated in separated studies. The impact of epigenetic inactivation was evaluated after methylation of their promoter and correlated to the outcome of metastasis in TNBC invasive cells. Demethylation treatments restoring the expression of target genes changed the progressive character of TNBC cell, allowing the definition of the promoter methylation impact on EMT. The identification of the signalling pathways impacted by this methylation enlightened on target gene functions and may lead to establish novel TNBC gene therapeutic strategies.

Finally, many studies have evaluated the role of DNA methylation in TNBC by inhibiting the regulatory sequences methylation of a number of specific genes. *CREB3L1* [128], *BRMS1* [129], *DACT2* [130], *WWC1* [131] and *ADAMTS18* [132] with anti-EMT activity were shown to be down-regulated in TNBC, while *FOXF2* [133] and *DKK2* [134] exerting pro-EMT activity were upregulated in TNBC; suggesting that DNA methylation is a leading source of the expression deregulation and TNBC high metastasis capacity (see Appendix A and Table 5).

## 6. Clinical Trials

Interpretation of clinical trials related to TNBC remains very complex due to the heterogeneous response to systemic therapy. In the last decade, multiple clinical trials have been conducted to control TNBC growth and metastasis, in order to define new targeted therapies and novel biomarkers. Furthermore, targeting PI3K, AKT, HDAC, PD-1, AR and Hsp 90 in metastatic TNBC with specific inhibitors and antagonists was the topic of a substantial number of clinical trials, including NCT01629615, NCT01790932, NCT02162719, NCT02353988, NCT02457910, NCT02657889, NCT02513472, NCT02768701, NCT02734290, NCT01349959, NCT02898207 and NCT02474173. Moreover, the effect of different drugs such as dasatinib, iniparib, epalrestat, tinostamustine and others was evaluated in various clinical studies NCT00371254, NCT01173497, NCT03244358, and NCT03345485, in the search of therapeutic biomarkers for metastatic TNBC.

Some authors have tried to revert the triple negative character of TNBC patients in order to give them a chance to benefit from treatments such as trastuzumab (Herceptin^®^) or tamoxifen (NCT01194908). It has been demonstrated that ER is not absent in some TNBCs but is epigenetically silenced by several histone and methyl groups leading to its inactivation. Thus, some clinical trials have intended to evaluate the effect of epigenetic reprogramming. Histone deacetylase inhibitors such as LBH589 and demethylating inhibitors can rid of histone- or methyl-group excess; the reactivation of ER opens the door to use hormonal agents in order to control the growth and the metastasis of TNBCs. Despite the number of metastatic TNBC clinical trials every year, drug resistance, wide heterogeneity and high metastasis rate keep TNBC treatment very challenging and epigenetic approaches must be considered and developed.

## 7. Conclusions

Triple negative breast cancer is a heterogeneous entity with a high rate of treatment failure due to the appearance of distant and/or nearby metastases. As a main factor responsible for this invasive capacity, EMT is an interesting prognostic biomarker and therapeutic target for TNBC patients. However, its clinical significance has only recently been investigated. Comprehensive new studies should shed light on EMT-related markers and particularly those implicated in epigenetic mechanisms. Thus, the study of epigenetic alterations in TNBC, encompassing lncRNAs, miRNAs, histone and DNA modifications, is a promising area to develop new treatments and EMT-related biomarkers. It is unlikely however that one epigenetic modification alone would be sufficient to identify robust and potential biomarkers of TNBC. A global study incorporating different epigenetic markers is necessary for understanding their role in TNBC. In conclusion, EMT is a promising anticancer target; its epigenetic alterations can be considered early biomarkers representing potential diagnostic and prognostic factors in order to define a global metastatic signature for TNBC.

## Figures and Tables

**Table 1 cancers-11-00559-t001:** Significant lncRNAs implicated in EMT of TNBC.

lncRNAs	Targets	Genes and Pathways Implicated	Sense of Dyregulation in TNBC	Role of LncRNA in TNBC	References
**LET**			Downregulated	Anti-tumoural	[21]
**XIST**	miR-155	CDX1	Downregulated	Anti-tumoural	[22]
**lncRNA-CTD-2108O9.1**	LIFReceptor	JAK/STAT and MAPK pathway	Downregulated	Anti-tumoural	[23]
**GAS5**	miR-196a-5p	FOXO1/PI3K/AKT pathway	Downregulated	Anti-tumoural	[24]
**MALAT1** **MALAT1/HuR (ELALV1) complex**	miR-1 miR-129-5p --	Slug -- CD133	Up-regulated Up-regulated Absence of complex	Pro-tumoural Pro-tumoural Anti-tumoural	[25] [26] [27]
**HOTAIR**			Up-regulated	Pro-tumoural	[28]
**SNHG12**		MMP13	Up-regulated	Pro-tumoural	[29]
**LINC01638**		c-Myc MTDH-Twist1 pathway	up-regulated	Pro-tumoural	[30]
**H19**	miR-675	ubiquitin ligase E3 family (c-Cbl and Cbl-b)	Up-regulated	Pro-tumoural	[31]
**Linc-ZNF469-3**	miR-574-5p	ZEB1	Up-regulated	Pro-tumoural	[32]
**LINC00673**		NCR3LG1(B7-H6)	Up-regulated	Pro-tumoural	[33]
**Lnc-ATB**	miR-141-3p	ZEB1 and ZEB2	Up-regulated	Pro-tumoural	[34]
**HOXA11-AS**			Up-regulated	Pro-tumoural	[35]
**OR3A4**			Up-regulated	Pro-tumoural	[36]
**MIAT** (*ceRNA*)	miR-155-5p	DUSP7	Up-regulated	Pro-tumoural	[37]
**GHET1**			Up-regulated	Pro-tumoural	[38]
**AC026904.1 and UCA** (*ceRNA*)		Slug	Up-regulated	Pro-tumoural	[39]
**NNT-AS1** (*ceRNA*)	miR-142-3p	ZEB1	Up-regulated	Pro-tumoural	[40]
**PVT1**		P21	Up-regulated	Pro-tumoural	[41]
**ARNILA** (*ceRNA*)	miR-204	Sox4	Up-regulated	Pro-tumoural	[42]
**LncRNA-RoR**	miR-145	ARF6	Up-regulated	Pro-tumoural	[43]
**LINC00518**		CDX2 and Wnt signaling pathway	Up-regulated	Pro-tumoural	[44]
**ADAM and lnc015192** *(ceRNA)*	miR-34a		Up-regulated	Pro-tumoural	[45]
**LOC554202**			Up-regulated	Pro-tumoural	[46]
**snaR**			Up-regulated	Pro-tumoural	[47]

**Table 2 cancers-11-00559-t002:** Major miRNAs involved in EMT of TNBC.

miRNAs	Targets and Pathways	Sense of Dyregulation in TNBC	Role of miRNA in TNBC	References
**miR-31**		Downregulated	Anti-tumoural	[46]
**DLK1-DIO3 region (miRs 300, 382, 494, 495, 539, 543, and 544)**	TWIST1, BMI1, ZEB1/2, and miR-200 family	Downregulated	Anti-tumoural	[61]
**miR200a**	ZEB1, ZEB2 and SUZ12 E-cadherin pathway EPH receptor A2 (EPHA2) EPHA2 pathway	Downregulated	Anti-tumoural	[62]
**miR200b**	PKCα/Rac1	Downregulated	Anti-tumoural	[63]
**miR-200c**	ZEB1	Downregulated	Anti-tumoural	[64]
**miR-30a**	Inhibition of ROR1	Downregulated	Anti-tumoural	[65]
**miR-146a**	Inhibition of RhoA	Downregulated	Anti-tumoural	[66]
**miR-143-3p**	LIMK1 expression LIMK1/CFL1 pathway	Downregulated	Anti-tumoural	[67]
**miR-200b-3p/miR-429-5p**	LIMK1 expression LIMK1/CFL1 pathway	Downregulated	Anti-tumoural	[68]
**miR-139-5p**	TGF-β, Wnt, MAPK, and PI3K	Downregulated	Anti-tumoural	[69]
**miR-212-5p**	Prrx2 Wnt/β-catenin pathway	Downregulated	Anti-tumoural	[70]
**miR-199a-5p**		Downregulated	Anti-tumoural	[71]
**miR-155**		Downregulated	Anti-tumoural	[72]
**miR-34c-3p**	MAP3K2 MAP3K2 pathway	Downregulated	Anti-tumoural	[73]
**miR-3178**	Notch1	Downregulated	Anti-tumoural	[74]
**miR-200b-3p**		Downregulated	Anti-tumoural	[75]
**miR-125b**	MAP2K7 MAPK pathway	Downregulated	Anti-tumoural	[76]
**miR-655**	Prrx1	Downregulated	Anti-tumoural	[77]
**miR27-a**	PTEN, PKB PI3K/AKT pathway	Up-regulated	Pro-tumoural	[78]
**miR-182**	Inhibition of FOXF2 PFN1	Up-regulated Up-regulated	Pro-tumoural Pro-tumoural	[79] [80]
**miR-454**	PTEN PI3K/AKT pathway	Up-regulated	Pro-tumoural	[81]
**miR-373**	HIF1α and TWIST HIF1α-TWIST pathway	Up-regulated	Pro-tumoural	[82]
**miR-221/222**	ADIPOR1 and TRPS1 NF-κB, IL6, and JAK2/STAT3 pathway	Up-regulated	Pro-tumoural	[83]
**miR-10b**	TGF-β1-induced EMT pathway	Up-regulated	Pro-tumoural	[84]

**Table 3 cancers-11-00559-t003:** Histone modification factors associated with EMT of TNBC.

Histones Modifications Factors	Targets and Pathways	Sense of Dyregulation in TNBC	Role of histone modifications in TNBC	References
**TIP60** *Lysine acetyltransferase*	Destabilize DNMT1 and inhibit SNAIL2 function leading to the inhibition of DNA methylation of EpCAM promoter region activating the expression of epithelial markers.	Downregulated	Anti-tumoural	[103]
**KDM6A** *H3K27me3-demethylase*	Maintenance of CDH1/E-cadherin levels. Role in the MET-associated resolution. Reactivation of bivalent genes by removing H3K27me3 marks deposited during EMT.	Downregulated	Anti-tumoural	[104]
**EZH2** *Histone-lysine N-methyltransferase (H3K27me3)*	Induce the repression of TIMP2 transcription increasing MMP-2 and MMP-9 activity.	Up-regulated	Pro-tumoural	[105]
**LSD1** *Lysine demethylase 1*	Represses E-cadherin expression by demethylating H3K4me at gene’s promoter, during which phosphorylation of LSD1 Ser112 is crucial.	Up-regulated	Pro-tumoural	[106]
**hSETD1A** *H3K4 Methyltransferase*	Activates MMPs expression (MMP2, MMP9, MMP12, MMP13, and MMP17).	Up-regulated	Pro-tumoural	[107]
**JMJD5** *H3K36me2 demethylase*	Promote cell invasion and induce EMT. Catalyze Snail promoter H3K36me2 demethylation activating its expression.	Up-regulated	Pro-tumoural	[108]
**KDM5B** *Lysine-specific demethylase 5B*	Overexpress MALAT1. Overexpress EMT markers, c-Met, Slug and N-Cadherin, and inhibitis E-Cadherin.	Up-regulated	Pro-tumoural	[109]
**KDM3A** *H3 lysine 9 demethylase 3A*	Promotes the expression of invasive genes by erasing H3K9me2 marks. Regulates the expression of MMP-9 and JUN expressions.	not evaluated	Pro-tumoural	[110]
**HDAC8** *Zinc-dependent class I HDAC*	Enhance TNBC cell migration. Regulates YAP protein levels by decreasing YAP phosphorylation at Ser127.	not evaluated	Pro-tumoural	[111]

**Table 4 cancers-11-00559-t004:** Role of histone modification inhibitors.

Treatment	Mechanisms	References
**Panobinostat (LBH589)** *Histone deacetylase inhibitor*	Increases CDH1 protein expression and morphology changes in MDA-MB-231 cells. Reverse EMT	[93]
**Suberoylanilide hydroxamic acid (SAHA) or Vorinostat** *pan-HDAC inhibitor*	Suppress metastasis by Inhibiting MMP-9 activity Promote EMT of TNBC cells via HDAC8/FOXA1 signals Downregulate E-cadherin Upregulate N-cadherin, vimentin and fibronectin	[94] [112]
**Entinostat (ENT)** *Class I HDAC (nuclear) inhibitor*	Increases E-cadherin transcription Reduces in N-cadherin mRNA expression Downregulates snail and twist Increases vimentin phosphorylation Increases vimentin filaments’s remodeling Reduces formation of tubulin-based microtentacles Upregulates cytokerain 8/18 Reverse EMT Reduce the percentage of TIC cells from TNBC cells. Reduce the CD44high/ CD24 low cell population, ALDH-1 activity, and protein and mRNA expression of known TIC markers such as Bmi-1, Nanog, and Oct-4.	[113] [114]
**Trichostatin A (TSA)** *HDAC inhibitor*	Reverse EMT Upregulate E-cadherin and suppresses Slug and Vimentin	[115]
**HTPB** *pan-HDAC inhibitor*	Inhibits integrin-b1/FAK/MMP/RhoA/F-actin pathways. Suppress tumour metastasis	[116]
**Parnate (PCPA)** *Lysine demethylase 1 LSD1 inhibitor*	Blockage of Slug/LSD1 interaction Induces the expression of E-cadherin and other epithelial markers	[117]
**UNC0638** *Histone methyltransferase inhibitor*	Restores E-cadherin and vimentin via modulating the Snail-G9a axis in order to block CSCs properties and reverse EMT.	[118]

**Table 5 cancers-11-00559-t005:** DNA modifications implicated in the induction of EMT.

Genes	Targets and Pathways	Sense of dyregulation in TNBC	Role of DNA methylation in TNBC	References
**CREB3L1** *Methylation of several key CpG sites*	Metastasis suppressor Activates p21 expression Suppresses cancer cell survival and angiogenesis genes.	Downregulated	Anti-tumoural	[128]
**BRMS1** *Promoter hypermethylation*	Metastasis suppressor. Downregulates several metastasis-related genes through modulating the activity of NF-kB, including osteopontin (OPN), urokinase-type plasminogen activator (uPA), micro-RNA-146, interleukin-6 (IL-6) and chemokine receptor 4 (CXCR4).	Downregulated	Anti-tumoural	[129]
**DAPPER2 (DACT2)** *Promoter CpG methylation*	Metastasis suppressor by antagonizing Wnt/β-catenin and Akt/GSK-3 signaling.	Downregulated	Anti-tumoural	[130]
**WWC1** *H3K27me3 inhibition catalysed by EZH2 and CpG island methylation mediated by DNMT1 within the wwc1 promoter*	Code for Kibra protein. EMT suppressor by regulating the Hippo/ YAP tumour suppressor pathway.	Downregulated	Anti-tumoural	[131]
**ADAMTS18** *Promoter methylation*	Deregulates AKT and NF-κB signaling, by inhibiting phosphorylation levels of AKT and p65.	Downregulated	Anti-tumoural	[132]
**FOXF2** *Absence of promoter methylation*	Functions as a promoter for EMT and metastasis. Regulates genes involved in controlling cell cycle progression and EMT, and is co-expressed with EMT genes SNAI2/Slug and Vimentin and a metastasis-promoting gene GLI2.	Up regulated	Pro-tumoural	[133]
**DKK2** *promoter CpG methylation*	EMT suppressor by suppressing canonical Wnt/β-catenin signaling via inhibiting β-catenin activity.	Downregulated	Anti-tumoural	[134]

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
