# Peer review of "New Insights into the Implication of Epigenetic Alterations in the EMT of Triple Negative Breast Cancer"

_cancers, 2019, doi:10.3390/cancers11040559_

Reviewer 1 Report

The review article by Noura Khaled and Yannick Bidet entitled “New insights into the implication of epigenetic alterations in the EMT of Triple Negative Breast Cancer” is well written, concise and complete. It is aimed at highlighting the epigenetic modifications correlated with EMT program. The subject of the manuscript is interesting and it is very important as understanding of this process may lead to development of novel targeting therapies in TNBC.

I have minor comments. I suggest to simplify the tables and add the page numbers missed in the references. Furthermore the following references regarding the role of EMT in TNBC and new biomarkers should be added:

- Hill BS et al Therapeutic Potential of a Novel αvβ₃ Antagonist to Hamper the Aggressiveness of Mesenchymal Triple Negative Breast Cancer Sub-Type. Cancers 2019;11(2). pii: E139.

- Camorani S, Inhibition of Bone Marrow-Derived Mesenchymal Stem Cells Homing Towards Triple-Negative Breast Cancer Microenvironment Using an Anti-PDGFRβ Aptamer. Theranostics. 2017;7(14):3595-3607      

Personally, I think that the manuscript is suitable for publication and that Cancers would be an appropriate place for it to be published.

Author Response

Response to Reviewer 1 Comments

We would like to thank the reviewer for careful and thorough reading of this manuscript and for the thoughtful comments and constructive suggestions, which help to improve the quality of this manuscript. 

The review article by Noura Khaled and Yannick Bidet entitled “New insights into the implication of epigenetic alterations in the EMT of Triple Negative Breast Cancer” is well written, concise and complete. It is aimed at highlighting the epigenetic modifications correlated with EMT program. The subject of the manuscript is interesting and it is very important as understanding of this process may lead to development of novel targeting therapies in TNBC.

Personally, I think that the manuscript is suitable for publication and that Cancers would be an appropriate place for it to be published.

Point 1: I suggest to simplify the tables and add the page numbers missed in the references.

Response 1: As suggested by the reviewer, we have simplified the tables, some columns have been removed or merged. The missing page numbers in the references have been added. Soma papers in online journals do not provide page numbers.

Point 2: Furthermore the following references regarding the role of EMT in TNBC and new biomarkers should be added:

- Hill BS et al Therapeutic Potential of a Novel αvβ₃ Antagonist to Hamper the Aggressiveness of Mesenchymal Triple Negative Breast Cancer Sub-Type. Cancers 2019;11(2). pii: E139.

- Camorani S, Inhibition of Bone Marrow-Derived Mesenchymal Stem Cells Homing Towards Triple-Negative Breast Cancer Microenvironment Using an Anti-PDGFRβ Aptamer. Theranostics. 2017;7(14):3595-360

Response 2: As suggested, we have added the 2 references regarding the role of EMT in TNBC and new biomarkers in the manuscript.

a. Regarding the role of EMT in TNBC:

“Inhibiting EMT by antagonizing the EMT-related mesenchymal cell markers such as αvβ3 integrin reveals its massive involvement in TNBC aggressiveness and metastasis.” Page 2 (line 49-51).

b. Regarding new biomarkers in TNBC:

“During the last decade, a large amount of data and knowledge were acquired on TNBC biomarkers. Chemokine C-C motif ligand 5 (CCL5), interleukin 17B, transforming growth factor β (TGF-β), placental growth factor, CXCL10 and platelet-derived growth factor receptor β (PDGFRβ) were considered as biomarkers for TNBC aggressiveness and progression. These biomolecules were involved in the recruitment of bone marrow-derived mesenchymal stem cells (BM-MSCs), which play a crucial role in multiple EMT program mechanisms (cell migration and invasion, neo-angiogenesis, promoting extracellular matrix remodelling...), towards TNBC sites.” Page 2 (line 53-59).

Reviewer 2 Report

This manuscript represents a concise review that covers the intersection of epigenetics, triple-negative breast cancer, and the epithelial-mesenchymal transition (EMT). It focuses on various epigenetic modifications - namely lncRNAs, microRNAs, histone modifications, and DNA methylation – that have been implicated to play in role in TNBC, EMT, or both.  The topic is timely and of interest to the field but the review should be improved prior to publication.

Major Points:

-It seems that many of the epigenetic alterations described relate to TNBC whereas others specifically relate EMT programs.  For the reader, this distinction is not always made clear.  And given the title and abstract, one would expect more of a focus on EMT in particular, rather than an overview of the slew on epigenetic and genetic changes that accompany transformation and tumorigenesis.  

-More than anything, the review presents a list of TNBC-associated lncRNAs, microRNAs, histone modifiers, and genes that are regulated by DNA methylation.  Much of this information is collected in tables that accompany the review.  At some level this tabulation is useful but it leaves a lot to interpreted and clarified by the reader.  It would be far more useful to describe in a bit more detail the mechanisms-of-action of some of the listed factors in order to highlight certain functional lessons that they teach.  In particular, such added emphasis could be placed on factors that specifically deal with EMT.  Indeed, a less comprehensive listing could actually be more informative than the long comprehensive lists of molecules that are included in these tables.

-In general, the section on histone modifications and DNA methylations is weaker than the lncRNA/microRNA sections. For the histone section, does expression of the various histone-modifying enzymes correlate with the histone modification profiles in TNBC?  For the DNA methylation section, a few descriptions of the downstream targets could be useful.  And once again, how these specifically modifications function during the EMT program is not clear.  

-It is very difficult to derive any meaning from Fig 1 without more of a description from the authors.  What are we looking at exactly?  Are these genes associated with activation of an EMT or are these changes that occur subsequent to/downstream of an EMT?  The tables are a bit more useful but some of the columns, e.g., EMT markers, seem repetitive and not very useful.

Minor Points:

-There are a few places in the text where the authors use an acronym that hasn’t been defined previously, e.g. p7, line 185 begins with “as mentioned” but I don’t believe the authors had discussed miR-200 yet. And on p15, line 293 the authors use BCSC but haven’t defined or described this term.  p7 line 158, RRA?

-It is unclear how the clinical trial section relates to the biochemical/molecular alterations discussed throughout the remainder of the text. 

Author Response

Response to Reviewer 2 Comments

We would like to thank the reviewer for careful and thorough reading of this manuscript and for the thoughtful comments and constructive suggestions, which help to improve the quality of this manuscript. 

This manuscript represents a concise review that covers the intersection of epigenetics, triple-negative breast cancer, and the epithelial-mesenchymal transition (EMT). It focuses on various epigenetic modifications - namely lncRNAs, microRNAs, histone modifications, and DNA methylation – that have been implicated to play in role in TNBC, EMT, or both.  The topic is timely and of interest to the field but the review should be improved prior to publication.

Major Points:

Point 1: It seems that many of the epigenetic alterations described relate to TNBC whereas others specifically relate EMT programs.  For the reader, this distinction is not always made clear.  And given the title and abstract, one would expect more of a focus on EMT in particular, rather than an overview of the slew on epigenetic and genetic changes that accompany transformation and tumorigenesis. 

Response 1: The structure of our review presents, that for each class of epigenetic factors (lncRNA, miRNA, histone or DNA modification), large screening studies that provide lists of dysregulated candidates in TNBC in general, in order to identify high impact candidate genes and factors. These candidates can relate to EMT but not exclusively. Thus, in the second part of each chapter, we focus on studies analysing the role of a specific epigenetic factor involved in the EMT of TNBC. The choice of the factors analysed often derived from the lists cited in the first paragraph. The functional evidences of each factor are not paraphrased in the text, but they are gathered in Tables 1 to 5. Few sentences at the end of Introduction try to make the structure more comprehensible (Page 2, lines 77-83).

Point 2: More than anything, the review presents a list of TNBC-associated lncRNAs, microRNAs, histone modifiers, and genes that are regulated by DNA methylation.  Much of this information is collected in tables that accompany the review.  At some level this tabulation is useful but it leaves a lot to interpreted and clarified by the reader.  It would be far more useful to describe in a bit more detail the mechanisms-of-action of some of the listed factors in order to highlight certain functional lessons that they teach.  In particular, such added emphasis could be placed on factors that specifically deal with EMT.  Indeed, a less comprehensive listing could actually be more informative than the long comprehensive lists of molecules that are included in these tables.

Response 2: Thanks to your first comment, we realised that the structure of the review was not clear to the readers. Our answer 1 should help on this point and, thus, make clear that all the epigenetic factors presented in the tables deal specifically with EMT and were identified as EMT enhancer or inhibitor factors. To facilitate and clarify the tables for the reader, a concise paragraph has been inserted in the manuscript at the end of the overview part of each epigenetic factor and before the paragraph of specific epigenetic factor in EMT presented in the table. Such a paragraph was already in the first version of the manuscript for lncRNAs. Please find below the sentences added for the other epigenetic factors.

miRNAs: Among the numerous miRNAs identified by these large screening assays, single studies aiming to investigate the precise performance of some specific miRNAs in the EMT of TNBC were proceeded. The goal was to determine the level of miRNAs expression in TNBC highly metastatic cells in vitro. and to alter this expression. The analysis of epithelial and mesenchymal markers was used to determine the anti- or pro-metastatic role of the candidate miRNAs. For a number of these miRNAs, target genes were predicted and tested in order to identify the specific mechanism of action of the miRNAs.

(Page 7-8, lines 224-231)

Histones modifications: Based on histone modifications profiling, several studies aimed to reconstitute the normal expression level of epigenetically deregulated genes. For this reason, metastatic TNBC cells have been treated with specific histone modifiers inhibitors to investigate their ability to induce or reduce TNBC metastasis and to evaluate their global effect on the TNBC EMT. Furthermore, the effect of depletion of upregulated histones modifiers on migration and invasive characteristics has been investigated in TNBC metastatic cells. Variations of epithelial and cell adhesion markers were also analysed in order to understand the roles of histone modifications on EMT and, eventually, identify potential diagnostic and prognostic biomarkers.

(Page 11-12, lines 290-297)

DNA methylation: Several genes chosen from the DNA-methylation-based profiles were evaluated in separated studies. The impact of epigenetic inactivation was evaluated after methylation of their promoter and correlated to the outcome of metastasis in TNBC invasive cells. Demethylation treatments restoring the expression of target genes changed the progressive character of TNBC cell, allowing to conclude about the impact of promoter methylation on EMT. The identification of the signalling pathways impacted by this methylation enlightened on target gene functions and may lead to establish novel TNBC gene therapeutic strategies.

(Page 15, lines 352-358)

Point 3: In general, the section on histone modifications and DNA methylations is weaker than the lncRNA/microRNA sections. For the histone section, does expression of the various histone-modifying enzymes correlate with the histone modification profiles in TNBC?  For the DNA methylation section, a few descriptions of the downstream targets could be useful.  And once again, how these specifically modifications function during the EMT program is not clear. 

Response 3: The sections on histone modifications and DNA methylation are indeed less plentiful than the lncRNA/microRNA sections. In contrast to the large number of studies over the last five years evaluating the role of lncRNAs and miRNAs in the EMT of TNBC, the literature is poorer about investigations on DNA methylation and histones modifications implications in the EMT of TNBC. However, targeting the molecular mechanisms of these two epigenetic factors could be an efficient strategy to prevent EMT in TNBC and we thought sections concerning these aspects were required. It might help promoting researches and gaining new insights in the topic.

The doubt about the role of all the histones methylation factors (H3K4me3, H3K27me3 and H3K36me3) mentioned in the table is linked to point 1 and 2 and the structure of the review. Tables always mention factors implicated in the EMT of TNBC. In TNBC, eight key histone modifications were identified, but to date only these three histone-modifying factors were verified and validated in single studies.

In response to the valuable comment about the lack of downstream targets of DNA methylation, a column was adjusted in the table to precise the main role of methylated genes. However, the downstream targets of these genes are not described. This lack of information reinforces the need to intensify the studies of these epigenetic factors in the EMT of TNBC. Some targets could indeed emerge as potential biomarkers and/or novel therapeutic targets.

Point 4: It is very difficult to derive any meaning from Fig 1 without more of a description from the authors.  What are we looking at exactly?  Are these genes associated with activation of an EMT or are these changes that occur subsequent to/downstream of an EMT?  The tables are a bit more useful but some of the columns, e.g., EMT markers, seem repetitive and not very useful.

Response 4: The epithelial-mesenchymal transition being a progressive and often a reversible process, setting start and end limits is a very subjective matter. Our choice is to include all the factors contributing to the morphological changes of the cells. In this aspect, although it could be debated, we have listed E- and N-Cadherin, Vimentin or Rho proteins, for example, as drivers of EMT and not downstream targets.

The aim of the Figure 1 is to assemble the factors cited in this review in order to build a bigger picture. As the first version was tough at first glance, we provide here an interactive version in which one can search, zoom and/or select items.

As suggested by the reviewer, all tables have been reduced to focus on compulsory information.

Minor Points:

Point 5: There are a few places in the text where the authors use an acronym that hasn’t been defined previously, e.g. p7, line 185 begins with “as mentioned” but I don’t believe the authors had discussed miR-200 yet. And on p15, line 293 the authors use BCSC but haven’t defined or described this term.  p7 line 158, RRA?

Response 5: The corrections have been made. Page 7, line 213: removed (as mentioned). Page 15, line 339: added “breast cancer stem cell (BCSC)”. Page 7, line 186: added “robust rank aggregation (RRA)”. Moreover, we have reviewed carefully the entire manuscript and have ensured the absence of further redundancies.

Point 6: It is unclear how the clinical trial section relates to the biochemical/molecular alterations discussed throughout the remainder of the text.

Response 6: The purpose of our review is not to explain the molecular mechanism of action of these epigenetic factors but rather to offer promising diagnosis and prognosis biomarkers along with novel therapeutic targets in the EMT of TNBC. Thus, a paragraph about ongoing clinical trials appears relevant. All clinical trials cited in our paper were conducted on metastatic TNBC patients in order to better understand and validate the significance of biomarkers in the EMT of TNBC. The biomarker-driven analyses of these cohorts help to validate biomarkers’ pertinence in TNBC via a specific targeting of EMT-related epigenetic factors.

Reviewer 3 Report

The authors gave a nice review of EMT-related epigenetic regulation in TNBC. I enjoy the read of the manuscript. A few suggestions are as follows:

1. EMT program is clear as starting from EMT inducing factors (hypoxia, TGFb, Wnts etc), to EMT promoting transcription factors (Snail, Slug, Twist, Zeb1/2 etc), to EMT-related functional markers (Ecad, Vim, MMPs etc). However, epigenetic factors are mostly regulating the downstream EMT markers. It would be nice to provide a clear indication of how the epigenetics was involved in the EMT program. Are there clues that epigenetic regulating enzymes are also targets of EMT-inducing factors or TFs? To avoid falling into the chicken/egg questions, is it possible the epigenetic regulation is running in parallel with EMT regulation? It will be nice to have authors’ opinions or perspectives.

2. As a global regulation of the whole genome/transcriptome, how the DNA/histone modifications were specified to a certain group of genes. In this case, how the global epigenetic changes impact differentially on epithelial and mesenchymal genes?

3. Fig 1. looks complex, but not very informative. It is hard to tell these small nodes. I would suggest listing the different epi-regulations, the groups of involved genes and their EMT-related targets out in a table.

4. Another topic which the authors may address a bit is whether the expression of miRNA and IncRNA were also regulated by epigenetic changes.

Author Response

Response to Reviewer 3 Comments

We would like to thank the reviewer for careful and thorough reading of this manuscript and for the thoughtful comments and constructive suggestions, which help to improve the quality of this manuscript. 

The authors gave a nice review of EMT-related epigenetic regulation in TNBC. I enjoy the read of the manuscript.

Point 1: EMT program is clear as starting from EMT inducing factors (hypoxia, TGFb, Wnts etc), to EMT promoting transcription factors (Snail, Slug, Twist, Zeb1/2 etc), to EMT-related functional markers (Ecad, Vim, MMPs etc). However, epigenetic factors are mostly regulating the downstream EMT markers. It would be nice to provide a clear indication of how the epigenetics was involved in the EMT program. Are there clues that epigenetic regulating enzymes are also targets of EMT-inducing factors or TFs? To avoid falling into the chicken/egg questions, is it possible the epigenetic regulation is running in parallel with EMT regulation? It will be nice to have authors’ opinions or perspectives.

Response 1: As suggested, we have added a paragraph giving a clearer indication of how the epigenetics is involved in the EMT program. It should also discuss the connection between the epigenetic factors and EMT markers both in term of facts and opinion.

“As researchers continue to illustrate the effect of epithelial-mesenchymal plasticity on tumour progression and to investigate its invasion-related molecular mechanism, a clearer perception of the general epigenetic modifications in malignant cells undergoing EMT is now appearing. Nevertheless, the current state of art of the precise role of epigenetic regulators in controlling both EMT and MET processes in tumorigenesis including TNBC are not fully understood yet. In sharp contradiction with EMT inducing factors and transcription factors, we are just now beginning to understand how those alterations induce EMT and MET or function to maintain epithelial and mesenchymal phenotypes. For this reason, the exact epigenetic regulation remains a frontier to discover.

We hypothesize that close interactions between different EMT inducing factors and TFs and epigenetic regulators exist. EMT-related factors and markers are targets of epigenetic regulations. Epigenetic modifiers must also be regulated by TFs to transcriptionally regulate epithelial and mesenchymal genes during EMT. For example, TFs may be responsible for the expression of epigenetic factors activating or repressing in return the expression of EMT-markers. The understanding of these probable functional interactions controlling the inter-conversion between epithelial and mesenchymal phenotypes may provide new opportunities and novel therapeutic approaches aiming towards the prevention of cancer invasion and metastasis, based on the development of individualized epigenetic cancer therapy.”

(Page 2, lines 69-86)

Point 2: As a global regulation of the whole genome/transcriptome, how the DNA/histone modifications were specified to a certain group of genes. In this case, how the global epigenetic changes impact differentially on epithelial and mesenchymal genes?

Response 2:  Studies have demonstrated that DNA methylation and histone modifications act in association to regulate genes. As an example, this cooperation helps to understand how DNA methylation can silence tumour suppressor genes and activate oncogenes and chromosome instability. At the level of each gene, the epigenome involves a combination of multiple events that is almost specific. CpG density or 3D nuclear position are examples of gene-specific components that drive or repress epigenetic marks. Depending on this combination, any alteration during tumourigenesis could modify expression both ways, as can transcription factors do, depending on their cofactors.

Previous gene-specific analyses of DNA methylation and histones modifications, as well as more recent genome-wide analyses, have unveiled a large panel of roles for each epigenetic modification individually. However, analysing their partnership is complex and it remains unclear which pattern of epigenetic alterations is critical for TNBC pathogenesis.

Point 3: Fig 1. looks complex, but not very informative. It is hard to tell these small nodes. I would suggest listing the different epi-regulations, the groups of involved genes and their EMT-related targets out in a table.

Response 3: The aim of the Figure 1 is to assemble the factors cited in this review in order to build a bigger picture. All nodes in the network already appear in one of the tables. As the first version was tough at first glance, we provide here an interactive version in which one can search, zoom and/or select items.

Technical comment to the editor: To build a dynamic figure people can play with online, Figure 1 is now in HTML5 format. Please find enclosed a zip archive containing a folder with all required files. It has been locally tested and works well on Firefox and Safari. Once hosted on a webserver it should work just as well on any modern navigator.

Point 4: Another topic which the authors may address a bit is whether the expression of miRNA and IncRNA were also regulated by epigenetic changes.

Response 4: This point completes the circle with point 1: miRNA and lncRNA have to be regulated in some way and epigenetics appears as a natural candidate. Some data are available, but they remain imprecise and molecular details are still lacking. To address this question, the following paragraph has been added: “Several data suggest that chromatin modifications are correlated with differential expression of lncRNAs and miRNAs in TNBC. Promoter hypermethylation was shown to be one of the major mechanisms to silence miR-200c/miR-141 and LCO554202/miR-31 locus. It is very likely that the same regulation occurs with other lncRNAs and miRNAs involved in the EMT of TNBC.” (Page 7, lines 221-224)